# Preference of CAMSAP3 for expanded microtubule lattice contributes to stabilization of the minus end

Hanjin Liu, Tomohiro Shima

**CAMSAPs are proteins that show microtubule minus-end–specific localization, decoration, and stabilization. Although the mechanism for minus-end recognition via their C-terminal CKK domain has been well described in recent studies, it is unclear how CAMSAPs stabilize microtubules. Our several binding assays revealed that the D2 region of CAMSAP3 specifically binds to microtubules with the expanded lattice. To investigate the relationship between this preference and the stabilization effect of CAMSAP3, we precisely measured individual microtubule lengths and found that D2 binding expanded the microtubule lattice by ~3%. Consistent with the notion that the expanded lattice is a common feature of stable microtubules, the presence of D2 slowed the microtubule depolymerization rate to ~1/20, suggesting that the D2-triggered lattice expansion stabilizes microtubules. Combining these results, we propose that CAMSAP3 stabilizes microtubules by lattice expansion upon D2 binding, which further accelerates the recruitment of other CAMSAP3 molecules. Because only CAMSAP3 has D2 and the highest microtubule-stabilizing effect among mammalian CAMSAPs, our model also explains the molecular basis for the functional diversity of CAMSAP family members.**

## Introduction

Microtubules are a eukaryotic cytoskeletal polymer with a hollow cylindrical structure composed of $\alpha$- and $\beta$-tubulin heterodimers (Zhang et al, 2018). They play essential cellular roles such as providing mechanical strength, building the machinery for cell division, and working as the railway on which motor proteins transport intracellular components (Dent & Bass, 2014; Goodson & Jonasson, 2018). Microtubules change their spatiotemporal distributions in cells in response to biochemical signals to construct the basic machinery for mitosis, cellular development, and migration (Williamson et al, 1996; Gadde & Heald, 2004; Dogterom & Koenderink, 2019). This dynamic reorganization is achieved mainly by two phenomena: polymerization/depolymerization and

nucleation. Polymerization and depolymerization refer to the addition/removal of tubulin dimers at preexisting microtubule tips, which results in microtubule elongation and shrinkage, respectively, a phenomenon usually called dynamic instability (Mitchison & Kirschner, 1984). On the other hand, microtubule nucleation generates new microtubules from tubulin dimers (Job et al, 2003; Ayukawa et al, 2021). The fine regulation of dynamic instability and nucleation is important for keeping microtubule networks ordered in cells and thus for proper cell activities.

Eukaryotes have a huge number of microtubule-associated proteins (MAPs) that alter microtubule dynamics. For instance, microtubule end–tracking proteins EB1 and XMAP215 increase the microtubule growth rate and catastrophe frequency, promoting microtubule dynamic instability (Zanic et al, 2013; Farmer et al, 2021), whereas cytoplasmic linker–associated proteins protect microtubules from depolymerization and suppress catastrophe (Aher et al, 2018). Among MAPs, calmodulin-regulated spectrin-associated proteins (CAMSAPs) and the invertebrate homolog patronin track the microtubule minus end and regulate microtubule dynamics (Hendershott & Vale, 2014; Jiang et al, 2014; Akhmanova & Steinmetz, 2019). Patronin/CAMSAPs are also hypothesized to be non-centrosomal microtubule nucleators, thus providing an indispensable role in cells that need microtubules to be nucleated far from the cell centrosome (Meng et al, 2008; Goodwin & Vale, 2010; Tanaka et al, 2012; Feng et al, 2019; Imasaki et al, 2022). In cells, mutations or knockdown of CAMSAPs disturb microtubule growth or patterning, reaffirming the importance of these proteins in microtubule organization (Tanaka et al, 2012; Toya et al, 2016; Pongrakhananon et al, 2018; Coquand et al, 2021).

Vertebrates generally have three CAMSAP members, CAMSAP1–3, which show slight differences in their interactions and functions with microtubules. In vitro experiments using mouse CAMSAPs found that CAMSAP1 can track the microtubule minus tip but does not affect microtubule dynamics. On the other hand, CAMSAP2 not only tracks the microtubule minus tip but also remains bound to the microtubule after minus-end elongation, resulting in microtubule decoration at the larger area of minus ends. It also stabilizes microtubules and protects microtubules from catastrophe. Finally, CAMSAP3 decorates and stabilizes microtubule minus ends, with a stronger effect than CAMSAP2. CAMSAPs consist of several coiled-

Department of Biological Sciences, Graduate School of Science, The University of Tokyo, Tokyo, Japan

Correspondence: tomohiro.shima@bs.s.u-tokyo.ac.jp

coil regions and a well-conserved C-terminal globular domain called CKK (Atherton et al, 2017), that is, responsible for the minus-end specificity of the proteins. Recent structural studies on the CKK domain have revealed how the CKK domain recognizes slight structural differences between microtubule minus ends and other regions (Atherton et al., 2017, 2019). However, these studies investigated the CKK domain only and failed to clarify the mechanism of the microtubule stabilization by CAMSAP2/3 and why CAMSAP1 does not have this property.

To understand the microtubule stabilization mechanism, it is necessary to consider the structure of not only CAMSAPs but also of microtubules. Recent cryogenic electron microscopy (cryo-EM), X-ray diffraction, and fluorescence microscopy studies have shown that microtubules can take diverse conformations depending on their binding states with nucleotides, proteins, and chemical agents. In terms of microtubule stability, the lattice expansion/compaction of the tubulin periodicity along the longitudinal axis shows a clear correlation. The inhibition of microtubule depolymerization by the very slowly hydrolysable nucleotide guanosine-5'-[($α$, $β$)-methyleno]triphosphate (GMPCPP), treatment with the anti-cancer drug taxol (Alushin et al, 2014; Kamimura et al, 2016), or mutations in the GTPase site (LaFrance et al, 2022) force microtubules to take conformations with the expanded lattice compared with the unstable GDP-bound state. Therefore, lattice expansion is thought to be a common conformational change needed for microtubule stabilization.

The lattice expansion/compaction provides a flexible way to control microtubule stability and is recognized by MAPs. The cell division master regulator TPX2 (Roostalu et al, 2015) and motor protein KIF5C (Nakata et al, 2011; Morikawa et al, 2015) can "read" the differences in microtubule conformations and preferentially bind to the expanded lattice. Interestingly, more recent studies have shown that both TPX2 and KIF5C actively introduce lattice expansion to compact microtubules or, in other words, "write" structural code on the microtubule and stabilize it upon their binding (Zhang et al, 2017; Peet et al, 2018; Shima et al, 2018). Considering the "reading" and "writing" abilities are conferred to KIF5C and TPX2 in a conjugative manner, it may be common for "reader" proteins to work as "writers" as well and stabilize microtubules through lattice expansion. Because preferential binding to the expanded lattice implies that the expanded lattice is energetically favorable for the "reader"–microtubule complex, this difference in free energy may be the driving force that expands the compact lattice.

The discovery of "reader/writer" MAPs raises the possibility that a similar "reader/writer" region exists in CAMSAP2/3 but not CAMSAP1. Truncation studies of CAMSAPs have demonstrated that CAMSAP2/3 have regions that, unlike the CKK domain, bind to the expanded lattice of GMPCPP-stabilized microtubule seeds but hardly bind to the compact lattice of dynamically polymerized GDP-bound microtubules (Jiang et al, 2014; Atherton et al, 2017). These truncated constructs have included the middle region, named microtubule-binding domain (MBD), or region adjacent to the CKK domain, named D2, for which little is known (see Fig S1 of Atherton et al, 2017). Because these regions seem to "read" the expanded lattice, we hypothesized they play a key role in stabilizing microtubules.

Here, we investigated the D2 region, which has 139 residues, of human CAMSAP3 for its contribution to microtubule stabilization. Our series of in vitro experiments revealed that D2 has a high affinity for various types of microtubules with the expanded lattice, including GMPCPP-stabilized, taxol-stabilized, and KIF5C-pretreated GDP-microtubules. Consistent with our hypothesis, an excess amount of D2 expanded the lattice of GDP-microtubules and inhibited depolymerization. These results strongly suggest that D2 shares similar lattice "reader/writer" abilities with KIF5C and, therefore, can regulate microtubule dynamics upon its binding. The synergy between the properties of D2 and the minus-end tracking ability of the CKK domain provides a common explanation for the complex physiological roles of CAMSAPs.

## Results

### D2 region of human CAMSAP3 preferentially binds to GMPCPP-stabilized microtubules

Previous in vitro microtubule dynamics assays have shown that the D2 region of mouse CAMSAP3 decorates microtubule seeds (Atherton et al, 2017). Because the microtubule seeds were composed of GMPCPP-bound tubulin, these results implied that the D2 region has a preference for GMPCPP-stabilized microtubules over standard GDP-bound microtubules (polymerized with GTP and hydrolyzed to GDP).

To verify if the same microtubule preference also holds for our D2 construct from human CAMSAP3, we conducted the binding assay using non-dynamic microtubules under total internal reflection fluorescence (TIRF) microscopy. The D2 region conjugated with fluorescent protein mCherry (D2-mCherry) was expressed and purified from *E. coli* using a histidine tag (Figs 1A and S1). GDP-bound microtubules (GDP-MT) were polymerized from GMPCPP-bound microtubules (GMPCPP-MT) immobilized on the glass surface by biotin/avidin conjugation (Fig 1B). In our assay, we did not immobilize GMPCPP-MT and GDP-MT separately for two reasons. First, when microtubules are spontaneously nucleated without seeds, most of the GMPCPP-MT (80–96%) forms a 14-protofilament conformation, whereas GDP-MT shows a wide distribution of protofilament numbers (Hyman et al, 1995; Rai et al, 2021). Therefore, to reduce the impact of the protofilament number, GDP-MT was polymerized from GMPCPP-MT to increase the ratio of 14-protofilament GDP-MT. Second, to analyze the structural transition of the GDP-MT lattice, we needed to eliminate the direct immobilization of GDP-MT on the glass surface because the biotin/avidin immobilization may compete with and prevent the longitudinal expansion and/or skewing of the GDP-MT lattice. We avoided this problem by indirectly immobilizing GDP-MT via GMPCPP-MT. Thus, using this assay system, we observed D2-binding on GDP- and GMPCPP-MT. As expected, D2 specifically decorated GMPCPP-MT (Fig 1C) but hardly bound to GDP-MT at D2 concentrations as high as 1 $μ$M. Quantification of the fluorescence images showed exclusive binding on GMPCPP-MT at 0.25 $μ$M D2 (GMPCPP-MT: 24 ± 6 a.u., GDP-MT: −0.6 ± 1.3 a.u.; mean ± S.D.) and 11-times higher affinity on GMPCPP-MT at 1 $μ$M D2 (GMPCPP-MT: 78 ± 16 a.u., GDP-MT: 7.2 ± 4.9 a.u.; mean ± S.D.)

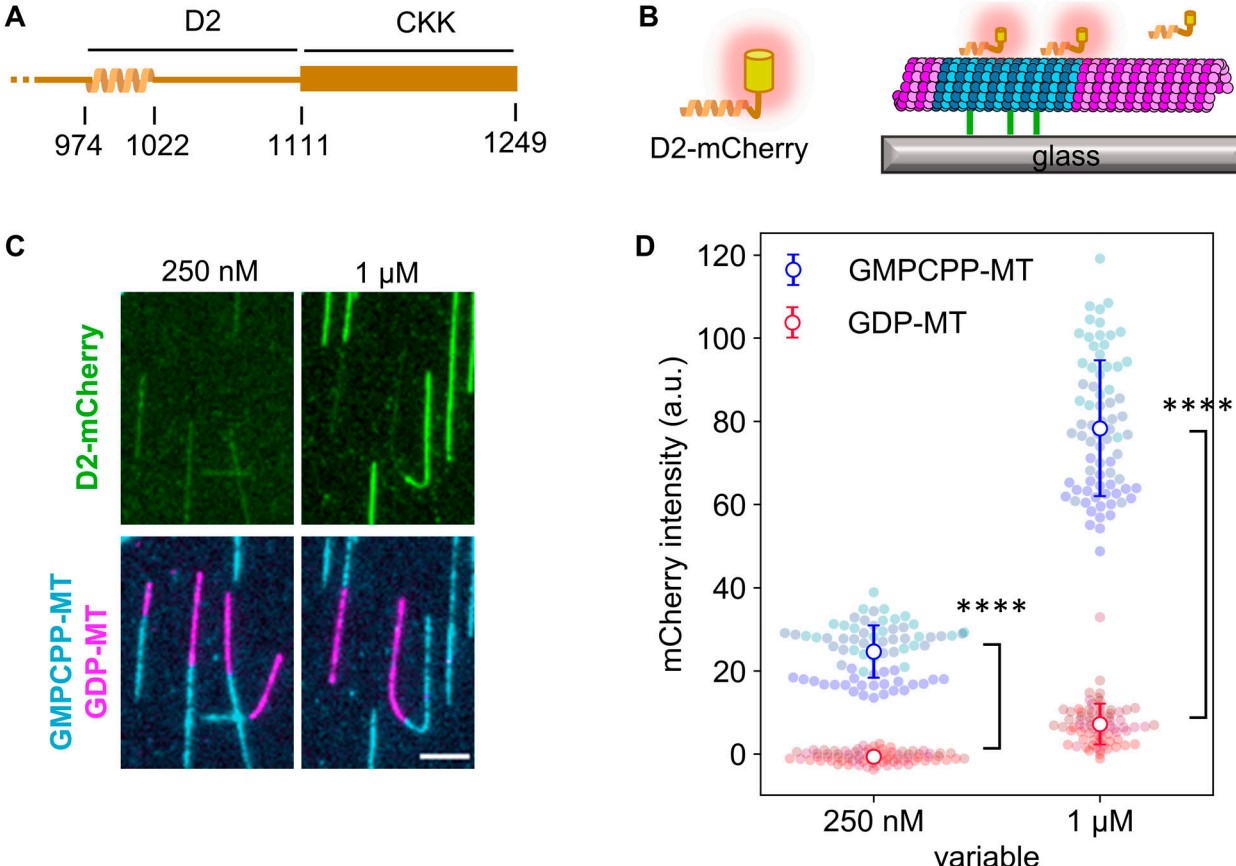

**Figure 1. D2 region in CAMSAP3 prefers GMPCPP-stabilized microtubules.**
**(A)** Schematic of human CAMSAP3 at its C-terminal. **(B)** Schematic of TIRF microscope used in the binding assay. The depolymerization of GDP-MTs was inhibited by adding 25% glycerol in the imaging buffer. We avoided using taxol in this assay because of its effects on the microtubule lattice. Blue indicates the GMPCPP-stabilized microtubule seed. Magenta shows GDP-MT elongated from the seed. **(C)** Representative images showing D2 binding to microtubules. Contrasts are set to the same range for each channel. Images after unmixing the fluorescence intensities are shown. Scale bar, 5 µm. 75 GMPCPP-MTs and 75 GDP-MTs from three replicates were analyzed for each D2 concentration. **(D)** Quantification of the average fluorescence intensity of D2-mCherry along microtubules. Colors represent three replicates. Error bars, S.D. ****$P < 0.0001$ (Welch's $t$ test).

(Fig 1D). Thus, we confirmed that D2 prefers GMPCPP-MT and is a common feature of mouse and human CAMSAP3.

### Affinity of D2 to microtubules increases in the presence of taxol

The preference of D2 to GMPCPP-MT is most likely due to the expanded lattice of GMPCPP-MT, in which the tubulin periodicity is expanded by 2–3% along the longitudinal axis compared with that of GDP-MT (Alushin et al, 2014). However, the different nucleotides in microtubules themselves (Estévez-Gallego et al, 2020) may also affect the affinity to D2. Therefore, we next investigated the affinity of D2 to other types of microtubules with expanded lattices. Taxol, a microtubule depolymerization inhibitor used as an anti-cancer drug, is known to expand microtubules and force microtubules to take a GMPCPP-bound–like structure (Yajima et al, 2012; Alushin et al, 2014; Kamimura et al, 2016). If the preference of D2 to GMPCPP-MT is attributed to the expanded lattice and not the interaction with the nucleotide itself, D2 should have a higher affinity for taxol-stabilized GDP-MT. To test this hypothesis, we conducted the same binding assay in the presence of 20 µM taxol, a concentration high

enough to force microtubules to take the expanded lattice (Yajima et al, 2012). Indeed, the presence of taxol increased D2-mCherry intensity on both GDP- and GMPCPP-MT, such that we could detect D2-binding on GDP-MT even at D2 concentrations as low as 100 nM (Fig 2A). Quantification revealed that the intensity of D2-mCherry on GDP-MT increased from an undetectable level to 22 ± 6 a.u. at 250 nM D2, which is similar to the intensity of D2 bound to GMPCPP-MT without taxol (Fig 2B). At 1 µM D2 with taxol, the amount of D2 bound to GDP-MT reached 85% that bound to GMPCPP-MT. This value remained constant (86%) when the D2 concentration was elevated to 4 µM, implying that the preference of D2 binding reached a plateau at 1 µM. These results support the idea that D2 distinguishes expanded/compact lattices, not the nucleotide state of microtubules.

We also tested the reversibility of the effects of taxol using a taxol-depletion assay (Fig 2A). In this assay, microtubules were first equilibrated with a solution containing D2 and taxol, and then taxol was subsequently depleted by exchanging the solution to D2 solution without taxol. Considering the fast dissociation ($k_{off}$ ~30 s$^{-1}$; Caplow et al,1994) and the weak affinity ($K_d$ ~2 µM; Li et al, 2000; Ross

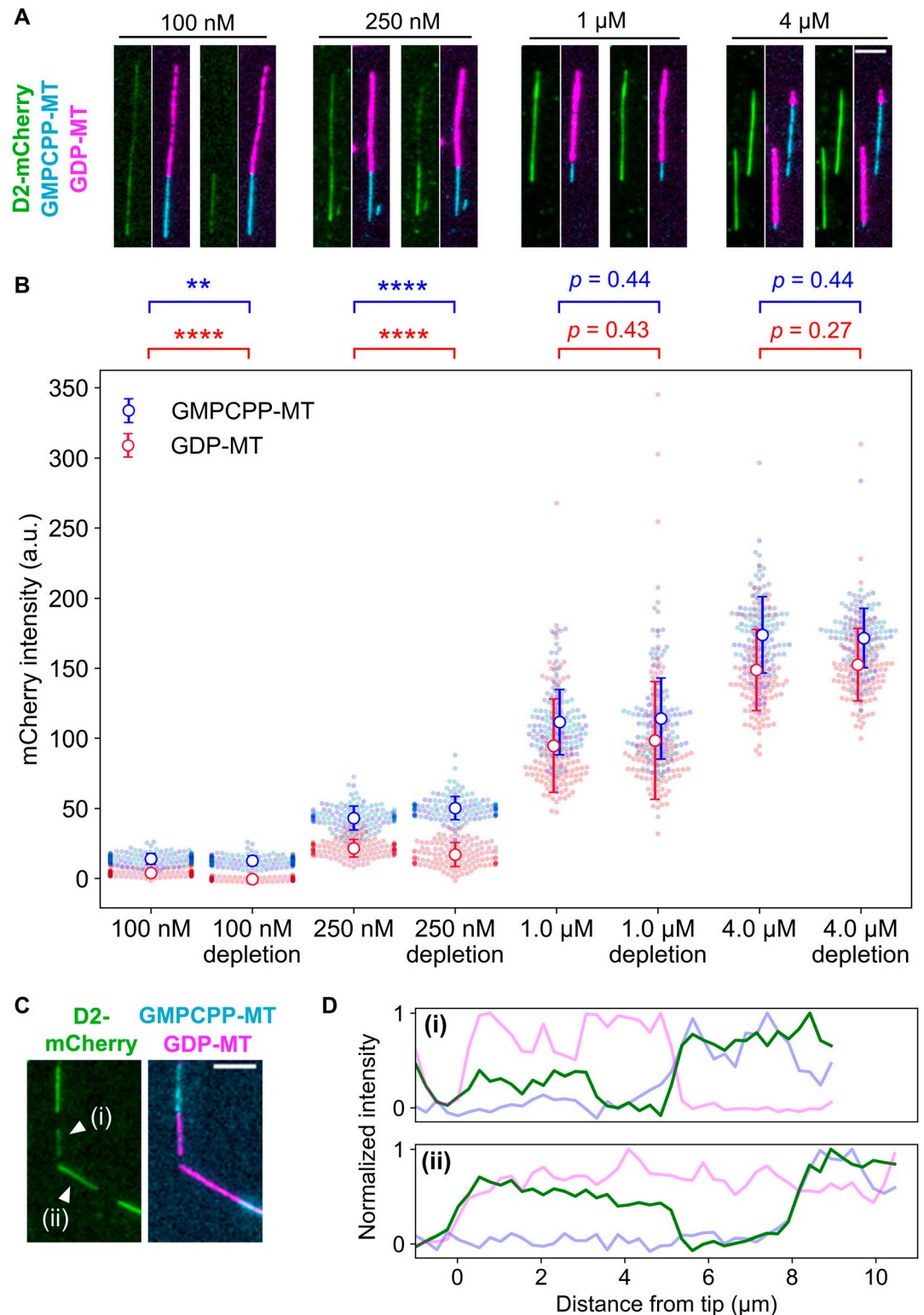

Figure 2. Affinity of D2 to GDP-MT was enhanced by the addition of taxol.

**(A)** Representative images of D2 binding to microtubules in the presence of taxol (left sides) or after 5-min taxol depletion (right sides). Contrasts were set to the same range for each concentration to highlight the difference before and after taxol-depletion. Scale bar, 5 μm. **(B)** Quantification of average fluorescence intensities of D2-mCherry. Error bars, S.D. **P < 0.01, ****P < 0.0001 (Welch's t test). Cohen's d after taxol depletion was d = 2.4 for 100 nM D2 and 0.60 for 250 nM D2. The mean value and sample size of each data set is summarized in Table 1. **(C)** Representative images of D2-rich subregions [marked as (i) and (ii)] on GDP-MT at 250 nM D2 after taxol depletion. Scale bar, 5 μm. **(C, D)** Line profiles of microtubules in (C) plotted against the distance from the tips.

**Table 1. mCherry intensity on microtubules in the taxol-depletion assay.**

| Condition | | mCherry intensity (a. u., mean ± S.D.) | | | |
|---|---|---|---|---|---|
| Microtubule | Taxol | 100 nM | 250 nM | 1 µM | 4 µM |
| GMPCPP-MT | 20 µM | 14 ± 4 (n = 150) | 43 ± 9 (n = 130) | 112 ± 23 (n = 130) | 173 ± 27 (n = 130) |
| | Depleted | 13 ± 3 (n = 150) | 50 ± 8 (n = 130) | 114 ± 29 (n = 130) | 171 ± 21 (n = 130) |
| GDP-MT | 20 µM | 4.0 ± 2.4 (n = 150) | 22 ± 6 (n = 130) | 95 ± 33 (n = 130) | 148 ± 29 (n = 130) |
| | Depleted | −0.5 ± 1.1 (n = 150) | 17 ± 8 (n = 130) | 98 ± 41 (n = 130) | 152 ± 25 (n = 130) |

et al, 2004) of taxol with GDP-MT, no more than ~1% of the bound taxol could remain bound or rebind during the incubation in taxol-free buffer. The presence of a stoichiometric amount of taxol is needed to expand the lattice (Kamimura et al, 2016). Therefore, after the taxol depletion, a subset of GDP-MTs was expected to take the compact lattice. At 1 µM or higher concentrations of D2-mCherry, the D2-mCherry fluorescence intensity on the MTs did not change after the taxol depletion. However, at lower D2 concentrations, the behavior was different, namely, at 100 nM D2, D2 molecules bound on the GDP-MT in the presence of taxol were completely dissociated after taxol depletion. Consequently, taxol depletion significantly decreased the D2 fluorescence intensity on GDP-MT to zero (4.0 ± 2.4 a.u. to −0.5 ± 1.1 a.u., Fig 2B and Table 1). At 250 nM D2, the D2 dissociation by taxol depletion was partial, but the intensity decrease was still significant (22 ± 6 a.u. to 17 ± 8 a.u., Fig 2B and Table 1). Under this condition, D2 occasionally remained on some subregions of GDP-MTs after taxol-depletion (Fig 2C), so that the line profiles along the microtubules clearly showed a stepwise increase in mCherry intensity (Fig 2D). The distinct amounts of residual D2 suggests that microtubules and/or D2 take distinct states in each subregion of GDP-MTs after taxol depletion. Together, these results demonstrate that D2 recognizes the microtubule lattice expansion, and changes in the affinity between D2 and microtubules are reversible at low D2 concentrations.

**D2 detects lattice expansion of native microtubules**

TIRF microscopy showed that D2 affinity for microtubule positively correlates with the microtubule lattice expansion induced by GMPCPP or taxol. However, these chemical reagents do not exist in intact mammal cells. To confirm whether D2 can identify microtubule conformations in normal cells, we induced microtubule lattice expansion using a protein, that is, present in the cytoplasm. The microtubule motor protein KIF5C is known to trigger the transition of the GDP-MT lattice structure from the compacted to expanded state (Peet et al, 2018; Shima et al, 2018). Importantly, the microtubule lattice remains expanded for ~2 min after KIF5C dissociation is induced by a high ionic strength buffer (Shima et al, 2018), but it is immediately compacted (<4 s) when buffer containing ADP or AMPPNP is used instead (Peet et al, 2018; Shima et al, 2018). Therefore, KIF5C treatment and washout are useful for modulating the lattice structure of native GDP-MT.

We repeated the KIF5C pretreatment assay with several concentrations of KIF5C and washout buffers whereas keeping the D2 concentration constant at 1 µM (Fig 3A). Compared with the case without KIF5C pretreatment, the D2-binding affinity to GDP-MT was

significantly increased by pretreatment with 0.1 µM KIF5C (Fig 3B). Pretreatment with higher concentrations of KIF5C further enhanced the D2 affinity for GDP-MT to almost the same level as that for GMPCPP-MT without the pretreatment (Fig 3B and C and Table 2). If D2 recognizes the lattice expansion induced by KIF5C, lattice compaction should lower the affinity between GDP-MT and D2 to the original weak state. Indeed, ADP-triggered lattice compaction upon KIF5C dissociation aborted D2 binding even at the highest KIF5C concentration tested (Fig 3B and C). The mCherry intensity on GDP-MT did not drop to the original value (P < 0.001, Steel–Dwass test) but resembled the condition with 0.1 µM KIF5C pretreatment. This incomplete rollback of the lattice structure may be due to the partial dissociation of D2 because ADP wash does not affect the binding of D2, which is a non-ATPase. To normalize the lattice structural deviation between conditions, we calculated the specificity of D2 binding by dividing the average mCherry intensity on GMPCPP-MT with that on GDP-MT (Fig 3D). This analysis revealed that the specificity of D2 monotonically decreased in response to large amounts of KIF5C pretreatment and restored the specificity to the original state after the ADP wash. This observation further supports the notion that D2 binding is sensitive to microtubule lattice spacing.

**D2 expands the microtubule lattice, but the expansion is not metastable**

Next, we speculated whether the "reader" ability of D2 contributes to expanding the lattice of GDP-MT when D2 excessively binds to it. To test this hypothesis, we conducted a microtubule expansion assay with a high concentration of D2 (Fig 4A). We polymerized Cy5-labeled GDP-MT from Alexa Fluor 488–labeled GMPCPP seed and capped it with Alexa Fluor 488–labeled GMPCPP-MT to prevent depolymerization during the assay. Before adding D2, microtubules were straight along their entire length. With 4 µM D2, the microtubules were severely deformed despite extensive parallel flow during the buffer exchange (Fig 4B). To precisely measure the microtubule length before and after D2 binding, we used a curve fitting algorithm on the fluorescence images. Microtubules were first fit to a two-dimensional (2D) B-spline Gaussian wall using the total fluorescence intensity. Subsequently, we measured the length of the Cy5-labeled region by a line scan of the Cy5 channel along the spline curve and fitting to the error function at both ends (see the Materials and Methods section). The presence of D2 caused a longitudinal elongation of GDP-MT that was associated with the lattice expansion. 4 µM D2 induced a 3.2 ± 0.8% (mean ± S.D.) lattice expansion, which is comparable to TPX2/GMPCPP–induced double

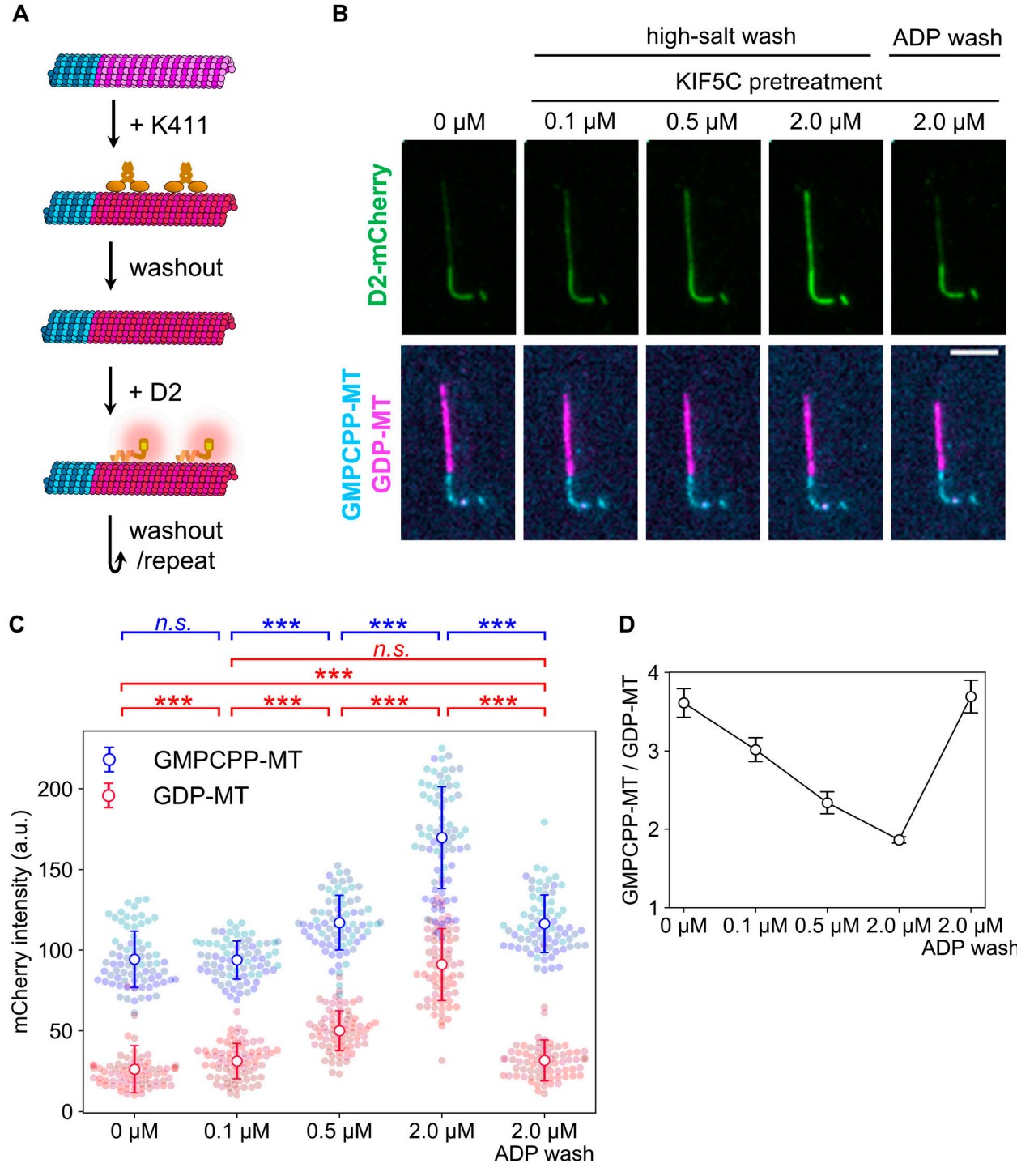

Figure 3. **Microtubule expansion by KIF5C pretreatment promotes D2 binding.**
**(A)** Overview of the KIF5C pretreatment assay. Coloring is the same as in Fig 1B except for the expanded GDP-MT lattice shown in red. KIF5C was added to induce microtubule expansion and washed out before applying D2. **(B)** Representative images after adding D2. Contrasts are set to the same values for each channel. Scale bar, 5 μm. **(C)** Quantification of D2-mCherry fluorescence intensity on microtubules. Colors indicate different replicates. Error bars, S.D. ***$P < 0.001$ (Steel–Dwass test); n.s., not significant. Sample size of each data set is summarized in Table 2. **(D)** Binding specificity of D2. Specificity was calculated as the mCherry intensity ratio on GMPCPP-MT to that on GDP-MT. **(C)** Error bars are the standard error predicted by pairs of standard errors in (C) and the error propagation rule.

expansion (Zhang et al, 2018), the most expanded state reported (LaFrance et al, 2022). Moreover, the addition of the CKK domain to the D2 construct decreased the expansion rate to 0.9 ± 1.2% (mean ± S.D., Fig S2), even though D2–CKK showed almost the same binding

**Table 2. mCherry intensity on microtubules in the kinesin pretreatment assay.**

| KIF5C concentration | mCherry intensity (a. u., mean ± S.D.) | | GMPCPP/GDP (mean ± S.E.) |
|---|---|---|---|
| | GDP-MT | GMPCPP-MT | |
| 0 µM | 26 ± 15 (n = 75) | 94 ± 17 (n = 75) | 3.61 ± 0.18 |
| 0.1 µM | 31 ± 11 (n = 75) | 93 ± 12 (n = 75) | 3.02 ± 0.15 |
| 0.5 µM | 50 ± 12 (n = 75) | 117 ± 17 (n = 75) | 2.34 ± 0.14 |
| 2 µM | 91 ± 22 (n = 75) | 170 ± 32 (n = 75) | 1.86 ± 0.04 |
| 2 µM (ADP wash) | 32 ± 13 (n = 75) | 116 ± 18 (n = 75) | 3.69 ± 0.21 |

distribution along microtubules as D2 at this concentration (Fig S2C and D). These results suggest that D2, not CKK, primarily leads the lattice expansion.

As mentioned above, the lattice expansion induced by KIF5C lasts for ~2 min even after KIF5C dissociation by a high ionic strength buffer. Therefore, the KIF5C-triggered expanded lattice is assumed to be a metastable state (Shima et al, 2018). Because D2 expanded the lattice more than KIF5C did, the D2-induced expansion may last longer than that induced by KIF5C. To check the metastability of the D2-induced expanded lattice, D2 was washed out with the high ionic strength buffer after the lattice expansion, and the microtubule length was measured. Contrary to our expectation, the lattice compaction was completed within 90 s after the D2 dissociation (Fig 4C). The results of the taxol-depletion assay (Fig 2), in which D2 dissociated from a subset of the microtubule region immediately after taxol removal, also support the notion that insufficient amounts of bound D2 cause an immediate return of the expanded lattice to the compact state. Note that in the KIF5C pretreatment assay (Fig 3), the enhanced D2 affinity remained after KIF5C dissociation by the same buffer. Therefore, the lattice expansion triggered by D2 is reminiscent of that by KIF5C, but their effects are different in terms of stability against the compaction. Finally, we confirmed no microtubule length dependency on the expansion and returning events (Fig 4D). Collectively, our expansion assay revealed that D2 can convert the microtubule lattice into the expanded state, but the D2-triggered expanded state is not metastable, so it returns to the compacted state immediately after the dissociation of D2.

### D2-triggered expanded lattice is resistant to microtubule depolymerization

We revealed that D2 binding transforms microtubules to the expanded state. Although many types of microtubules with the expanded lattice are resistant to depolymerization, it is unclear if this is true for D2 induction. Therefore, we quantified the depolymerization rate by glycerol depletion with or without 4 µM D2 (Fig 5A). A comparison of time lapse images in the absence (Fig 5B) and presence of D2 (Fig 5C) clearly indicated that depolymerization was strongly suppressed by 4 µM D2. Representative kymographs also showed an obvious difference between the two conditions (Fig 5D and E). Before quantifying the depolymerization rate, we could not build kymographs for most microtubules because microtubules changed their curves during depolymerization. As a workaround, we derived the depolymerization rate from the total fluorescence intensity in each specific region that surrounds individual

microtubules throughout the imaging time (Fig S3). We found that microtubule depolymerization dramatically slowed down from $0.82 ± 0.30$ $\mu$m s$^{-1}$ (mean ± S.D.) to $0.046 ± 0.028$ $\mu$m s$^{-1}$ upon exposure to 4 $\mu$M D2 (Fig 5F). We also investigated if D2 accumulates at the depolymerizing microtubule tips like MAPs such as Ska1 complex (Schmidt et al, 2012) because tip accumulation will expand microtubules more and effectively prevent further depolymerization. However, we found no D2 localization, as no intensity peaks were detected in the averaged line scans (Fig S4). These experiments suggest that the lattice expansion by D2 protects microtubules from depolymerization.

## Discussion

In this study, we demonstrated that the D2 region of human CAMSAP3 preferentially binds to the expanded lattice of microtubules. At saturating levels, D2 expanded the microtubule lattice by 3.2% (Fig 4), which slowed the microtubule depolymerization rate down ~18-fold (Fig 5). Combining these results with previously known functions of the CKK domain, we suggest a detailed model of microtubule stabilization by CAMSAP3 (Fig 6A). The interaction between microtubules and CAMSAP3 is first formed via the CKK domain because it occurs regardless of the presence of other CAMSAP3 domains (Jiang et al, 2014). Because D2 alone does not track either end of dynamic microtubules (Atherton et al, 2017), this CKK binding is likely needed for D2 to bind at the minus end of microtubules. Once a CAMSAP3 protein is bound to the microtubule surface, the local concentration of the D2 region at the minus end is dramatically increased. Assuming that the peptide between the microtubule binding region of D2 and the CKK domain (residues 1,023–1,111 for human CAMSAP3) takes the disordered conformation, which is predicted by AlphaFold2, and the inter-residue distance is 0.4 nm (Ainavarapu et al, 2007), a D2 molecule is confined to a volume of $4/3 × \pi × (0.4 × 88)^3 = 9.1 × 10^4$ nm$^3$ after CKK binding; thus, the D2 local concentration is estimated to be over 20 $\mu$M. Note that the local concentration is likely underestimated because the peptide chain is unlikely to be tautly stretched because of entropic energy loss. Considering 4 $\mu$M, D2 was sufficient to expand the lattice of GDP-MT (Fig 4B), a local concentration over 20 $\mu$M should be high enough to convert the minus end into the expanded state with high affinity for D2. As a result, CAMSAP3 protein strongly interacts with the minus end of microtubules via divalent interactions of the CKK domain and D2 region. At the same time, the minus end is stabilized by the D2-induced lattice expansion (Fig 5F).

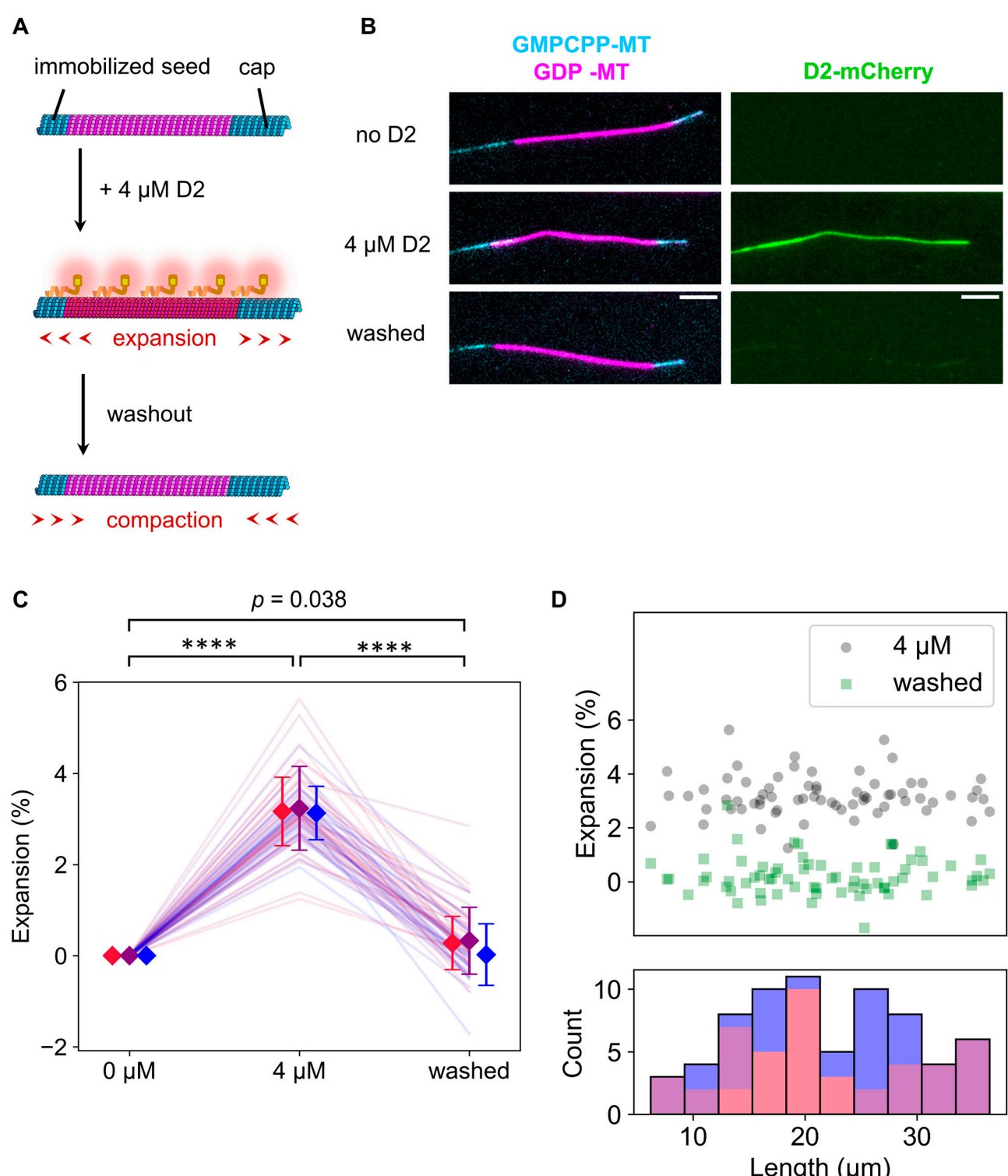

**Figure 4. Microtubule lattice expansion triggered by D2 binding.**
**(A)** Schematic illustration of the assay. Coloring is the same as in Fig 3A. **(B)** Representative images during the expansion assay. Red-to-green and green-to-blue fluorescence leakage were unmixed. Scale bar, 5 $\mu m$. **(C)** Quantification of the normalized microtubule length between Alexa Fluor 488–labeled GMPCPP seed and Alexa Fluor 488–labeled GMPCPP-capped microtubules. Microtubule length was calculated by 2D Gaussian fitting of the total intensity and spline fitting in the Cy5 channel. Each color represents a different replicate. Error bars indicate mean and S.D. of each replicate. ****$P < 0.0001$ (paired $t$ test with Bonferroni correction). 69 microtubules from three replicates were analyzed. **(D)** Scatter plot between the expansion rate and the microtubule lengths (upper) and a stacked histogram of the lengths under the 0 $\mu M$ D2 condition (bottom). **(C)** Each color corresponds to the same replicate in (C).

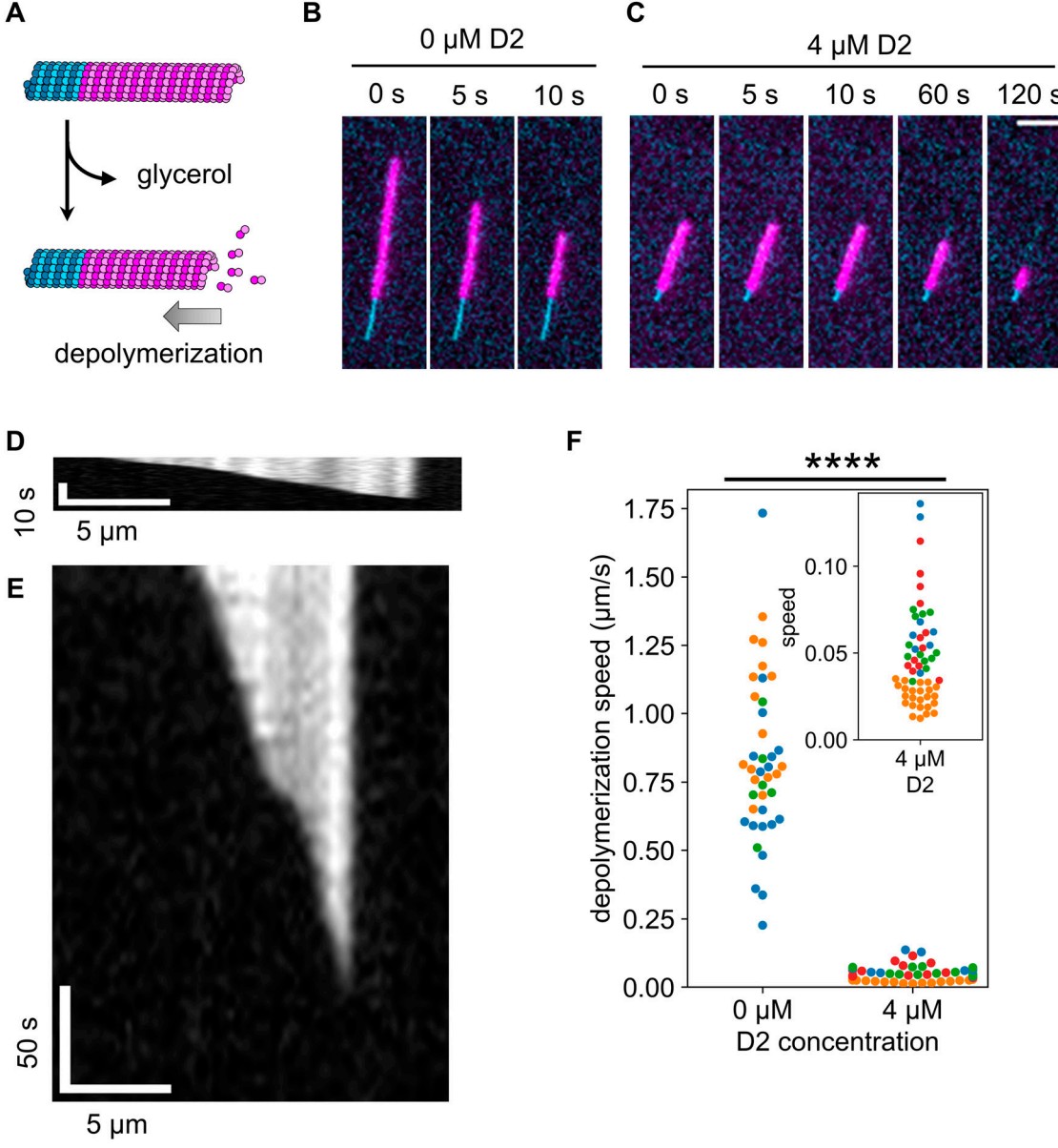

**Figure 5. Microtubule depolymerization was suppressed by D2 binding.**
**(A)** Schematic illustration of the assay. Microtubule depolymerization was induced by replacing the buffer with glycerol-free buffer. **(B, C)** Representative frames during depolymerization without D2 (B) or with 4 μM of D2 (C). The cyan channel image (GMPCPP seed) was taken only at the first frame so that the same image was used for each frame. Scale bar, 5 μm. **(B, C, D, E)** Kymographs of the microtubules in (B, C), respectively, depolymerizing without D2 (D) or with 4 μM D2 (E). Because the images were taken at different time intervals, two kymographs were resized to fit the same time scale. **(F)** Microtubule depolymerization speed. Different colors represent different replicates. 0 μM: 0.82 ± 0.30 μm/s (mean ± S.D., 40 microtubules, three replicates); 4 μM: 0.046 ± 0.028 μm/s (mean ± S.D., 56 microtubules, four replicates). ****P < 0.0001 (Welch's t test).

This expanded lattice recruits more CAMSAP3 molecules by promoting D2-microtubule interactions and prevents the dissociation of CAMSAP3. Previous work using recombinant CAMSAPs also supports the importance of non-CKK domains on the enhancement of minus-end affinity (Hendershott & Vale, 2014). Newly polymerized microtubule minus ends should interact with other CAMSAP3 molecules in the cytoplasm via the CKK domain, resulting in microtubules near the minus end decorated with CAMSAP3 over a long range (Jiang et al, 2014).

This model is consistent with the correlation between the domain composition of vertebrate CAMSAPs and their ability to decorate and stabilize microtubules. Previous truncation studies have shown that the D2 domain of CAMSAP3 and MBD in CAMSAP2/3 preferentially bind to GMPCPP-MT over GDP-MT, implying that these regions recognize the expanded lattice (Jiang et al, 2014; Atherton et al, 2017). Similar to the number of regions showing this preference, the microtubule-stabilizing effect is higher for CAMSAP3 than CAMSAP2. Also, our model can explain why CAMSAP1, which does not

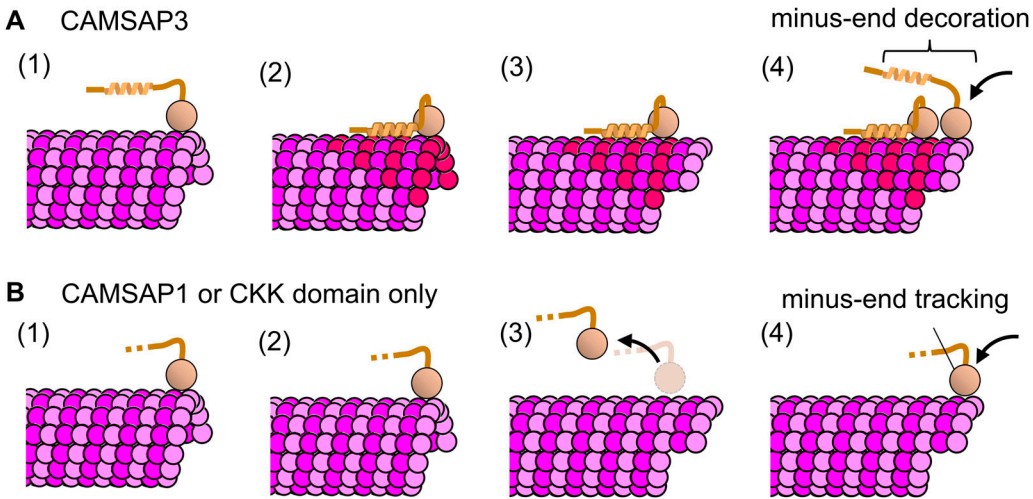

**Figure 6.  Model of minus-end decoration and stabilization by CAMSAP3 and comparison with CAMSAP1.**
**(A)** CAMSAP3 can specifically bind to the microtubule minus-end owing to its CKK domain (1), and the adjacent D2 region expands the microtubule lattice (2). The strong interaction between D2 and expanded microtubule (red tubulins) keeps CAMSAP3 molecules bound on the microtubule surface even after the microtubule is polymerized (3). The newly formed minus end provides additional binding sites for other CAMSAP3 proteins (4). This mechanism was observed as minus-end "decoration" under TIRF microscopy. **(B)** CAMSAP1 can specifically bind to the microtubule minus-end (1) but has no effect on the microtubule lattice structure (2). Upon polymerization (3), CAMSAP1 dissociates from the microtubule because where it bound is no longer a minus end. The newly formed minus end is open to CAMSAP1 proteins so that CAMSAP1 appears to be "tracking" the minus end.

have any region with a high affinity for the expanded lattice, barely affects microtubule stability (Fig 6B). Thus, the number of regions that prefer the expanded lattice is apparently proportional to the microtubule stabilization effect (Hendershott & Vale, 2014; Jiang et al, 2014). In this study, we focus on the D2 region of CAMSAP3 because of difficulty in expressing MBD in soluble fractions in our system. Future analysis of MBD together with mutational and swapping analyses of these regions will strengthen our model and provide a comprehensive understanding of the roles of MBD and D2.

Overall, we demonstrated that the D2 region of CAMSAP3 has two functions: to identify the expanded microtubule lattice (Figs 1–3) and to expand the compact lattice of GDP-MT (Figs 4 and 5). Both of these functions are common with the motor protein KIF5C and mitosis regulator TPX2. If we assume that the "reading" and "writing" codes on the lattice are correlated, there is a trade-off between the binding specificity of the proteins and their effectiveness at expanding the lattice. If the specificity is low (low reading ability), like in the case of KIF5C, the protein can spontaneously bind to the compacted lattice well, but the resulting expansion is small (low writing ability). Indeed, KIF5C expands the lattice by only ~1% (Peet et al, 2018; Shima et al, 2018), whereas D2 expands it by ~3% (Fig 4C). On the contrary, proteins with high specificity, like TPX2 and D2, are expected to expand the lattice longer (high writing ability) but do not bind well on a compact lattice because of the higher specificity for the expanded condition. Therefore, these highly specific proteins require another mechanism to recruit proteins on a compacted lattice before the expansion occurs. As for TPX2, recent light and electron microscopy studies have shown that it is locally concentrated on microtubules by liquid–liquid phase separation and Rayleigh instability (King & Petry, 2020; Setru et al, 2021). As for D2, the CKK domain in the same CAMSAP3 molecule is most likely concentrated at the microtubule minus-end (Fig 6A). Thus, the "reading" and "writing" abilities of these proteins may have been evolutionarily optimized according to their protein structure, binding partners, and cellular distribution and function.

Having both "reading" and "writing" ability can produce positive feedback between the binding affinity and the lattice structure of microtubules, resulting in the cooperative binding/dissociation of MAPs. KIF5C, for instance, shows cooperative binding on GDP-MTs so that it accumulates in a subset of GDP-MTs, whereas other subsets bound to very few KIF5C molecules (Shima et al, 2018). D2 demonstrated similar cooperativity in our taxol-depletion assay. At 250 nM D2 after taxol depletion, GDP-MTs showed two distinct types of affinity to D2: subregions where many D2 remained bound and where most D2 are dissociated (Fig 2C and D). Because of the similarity with KIF5C, we assume that this cooperative dissociation of D2 is a consequence of the positive feedback of D2 dissociation and lattice compaction. When a number of D2 molecules exceeding a threshold stochastically dissociated from a certain subregion of GDP-MTs, the lattice of the subregion should return to the compacted form, which lowers the affinity to D2, resulting in further D2 dissociation. Because microtubule lattice spacing is interconnected with adjacent tubulin dimers, tubulin dimers in a certain region of microtubules are likely to have similar lattice spacing as observed in Fig 2C. However, the cooperativity of D2 binding/dissociation was only seen under a specific condition. Thus, the positive feedback between the binding affinity and the lattice structure is likely less in the case of D2 compared with other MAPs that have "reading" and "writing" abilities.

Under low concentrations of D2 (100 and 250 nM), the D2 intensity on GMPCPP-MTs and GDP-MTs significantly changed after taxol depletion (Fig 2B). In all cases except on GMPCPP-MTs at 250 nM D2, the D2 intensity was decreased by taxol depletion. These results are

consistent with taxol depletion reversibly changing the MT lattice into a compacted form, which decreases the affinity with D2. However, in the case of GMPCPP-MTs at 250 nM D2, the D2 intensity increased by ~16% after taxol depletion (Table 1). This effect may be due to the increase in total D2 concentration. In our assays, taxol was depleted by introducing D2 solution without taxol. Therefore, because of remaining D2 molecules on microtubules, the total D2 concentration in the chamber may be higher than 250 nM after taxol depletion. This situation is also true for other concentrations, but the strength of the effect depends on the amount of remaining D2 on microtubules. At 250 nM D2, D2 showed frequent binding on GMPCPP-MTs but not on GDP-MTs in the absence of taxol (Fig 1C and D); thus, more D2 should remain bound on GMPCPP-MTs than on GDP-MTs after taxol depletion. Therefore, the effect of taxol depletion can be explained by the competing effects of residual D2 and changes in the lattice structure. These competing effects are insignificant when most of the binding sites on MTs are occupied by D2, which corresponds to fewer binding sites and an expanded lattice, as shown in assays using higher D2 concentrations (Fig 2, 1 and 4 μM).

It is worth noting that the microtubule structure is more complex than just an expanded or compact lattice. Contrary to KIF5C binding, the lattice expansion triggered by D2 binding was not metastable, returning to the original status immediately after D2 dissociation (Fig 4C). Notably, KIF5C-induced expansion lasts longer even though the relative expansion rate is less than that induced by D2, highlighting the differences in KIF5C- and D2-induced expanded lattices. There is also an inconsistency in that D2 binding, and GMPCPP-MT was enhanced by KIF5C pretreatment (Fig 3) even though KIF5C does not expand the lattice of GMPCPP-MT further (Shima et al, 2018). Also, even though the GTP-cap of microtubules is considered to have an expanded lattice structure (Zhang et al, 2018), one study showed that D2 does not track either end of dynamic microtubules, suggesting that D2 does not prefer the lattice of the GTP-cap (Atherton et al, 2017). Future structural studies on D2-coated microtubules and comparisons with the structure of KIF5C-bound GDP-MT (Shima et al, 2018) or the GTP-cap of dynamic microtubules will reveal the structural diversity of microtubules with expanded lattices and explain the causes for these differences.

## Materials and Methods

### Protein purification and labeling

Tubulin was purified from porcine brains by polymerization/depolymerization cycles in high molarity buffer as described previously (Castoldi & Popov, 2003). For GTP-bound tubulin, polymerization/depolymerization was repeated for three cycles. In the final cycle, tubulin was depolymerized in PEM (100 mM PIPES-KOH, 1 mM EGTA, and 2 mM $MgSO_4$, pH 6.9) supplemented with 0.1 mM GTP, snap-frozen, and stored in liquid nitrogen. To prepare GMPCPP-bound tubulin, two cycles were conducted in the same way as GTP-bound tubulin, and purified tubulin was subsequently subjected to two polymerization/depolymerization cycles using 0.1 mM GMPCPP (NU-405L; Jena Bioscience) instead of 1 mM GTP. In

the final cycle, tubulin was depolymerized in PEM containing 0.1 mM GMPCPP and stored in the same way as GTP-bound tubulin.

Fluorescently labeled tubulin and biotinylated tubulin were prepared by adding Alexa Fluor 488 NHS ester (A20000; Thermo Fisher Scientific), sulfo-Cy5 NHS Ester (13320; Lumiprobe), or sulfo-Cy5 NHS Ester (13320; Lumiprobe) dissolved in DMSO to polymerize microtubules. Unreacted labels were removed by two dilutions and ultracentrifugation of the microtubules. Labeled tubulin samples were mixed with intact tubulin at the indicated labeling rates, snap-frozen, and stored in liquid nitrogen.

The human *CAMSAP3* gene was bought from RIKEN (GNP Clone IRAL050D13) and cloned into pET11a plasmid vector. Plasmids were transformed into Rosetta II competent cells (71397; Novagen) and incubated in 2× YT medium (16 g/liters bacto-tryptone, 10 g/liters yeast extract, and 5 g/liters NaCl) supplemented with 10 μg/ml ampicillin in a 37°C shaker until the OD600 reached ~0.6. Protein expression was induced by 5.5-h incubation at 18°C in the presence of 0.5 mM IPTG. Bacteria were collected, snap-frozen in liquid nitrogen, and stored at −80°C. For protein purification, bacteria pellets were resuspended in lysis buffer (50 mM Tris–HCl, 500 mM NaCl, and 10 mM imidazole, pH 8.0) supplemented with protease inhibitor cocktail (1 mM 4-(2-aminoethyl) benzenesulfonil fluoride, 20 μM Leupeptin, 8.7 μM pepstatin, and 1 mM N-p-tosyl-L-argininie-methyl ester), sonicated, and pelleted by centrifugation at 18,000g for 10 min. D2-mCherry was retrieved from the supernatant using its 6× histidine tag by Ni-NTA agarose resin (141-09764; FUJIFILM). The resin was washed twice with 1.5 column volumes of wash buffer A (50 mM Tris–HCl, 500 mM NaCl, and 100 mM imidazole, pH 8.0), twice with 1.5 column volumes of wash buffer B (50 mM Tris–HCl, 500 mM NaCl, and 150 mM imidazole, pH 8.0), and finally once with wash buffer C (50 mM Tris–HCl, 500 mM NaCl, and 300 mM imidazole, pH 8.0). The second fraction from the washout using wash buffer B was concentrated and mixed with BRB30 (30 mM PIPES-KOH, 1 mM EGTA, and 2 mM $MgSO_4$, pH 7.0) using a 30-kD cutoff centrifugal filter (UFC503096; Merck). Finally, D2-mCherry solution was snap-frozen with liquid nitrogen and stored at −80°C.

KIF5C (residues 1–411) was purified from the *E coli* recombinant protein purification system. Mouse *Kif5c* gene was cloned into pET11a plasmid with 6xHis-tag inserted in the N-terminal end. Plasmids were first transformed into Rosetta II competent cells and incubated in 2× YT medium supplemented with 10 μg/ml ampicillin and 10 μg/ml chloramphenicol in a 37°C shaker until the OD600 reached 0.36. Protein expression was induced by a 14-h incubation at 18°C in the presence of 0.25 mM IPTG. Bacteria were resuspended in lysis buffer (PEM, 150 mM NaCl, and 10 mM imidazole, pH 6.9, supplemented with protease inhibitor cocktail), sonicated, and pelleted by centrifugation. KIF5C was incubated with Ni-NTA agarose resin and washed with 1.5 column volumes of wash buffer A (PEM, 150 mM NaCl, and 100 mM imidazole, pH 6.9) and twice with 1.5 column volumes of wash buffer B (PEM, 150 mM NaCl, and 150 mM imidazole, pH 6.9). KIF5C was eluted from the resin with elution buffer (PEM, 150 mM NaCl, and 300 mM imidazole, pH 6.9). The elution fraction was concentrated and buffer exchanged to BRB80 buffer (80 mM PIPES-KOH, 1 mM EGTA, and 2 mM $MgSO_4$, pH 7.3) supplemented with 8% glycerol and 0.5 mM DTT using a 30-kD cutoff centrifugal filter (UFC503096; Merck). Finally, KIF5C solution was snap-frozen with liquid nitrogen and stored at −80°C.

## Preparation for binding assay

A PEG-biotin–coated glass chamber was made by silane coupling and a succinimidyl ester reaction. Cover glasses (C022221S; Matsunami Glass) were first subjected to sonication in 1 N KOH and plasma treatment. Glass surfaces were coated with an amino group by sandwiching N-2-(aminoethyl)-3-aminopropyl-triethoxysilane (KBE-603; Shin-Etsu Chemical) and incubation for 20 min at room temperature. These glasses were washed with deionized water 20 times and then incubated with 200 mg/ml of 0.5% biotinylated NHS-PEG (ME-050-TS and BI-050-TS; NOF) for 90 min at room temperature. Coated glasses were stuck to each other with 30-$\mu$m double-sided tape (5603; Nitto-Denko) to make an ~2-mm wide flowing chamber. The glasses were sealed in a food saver in vacuo and stored at –80°C until use.

GMPCPP-bound tubulin labeled with Alexa Fluor 488 (2% labeled) and biotin (5% labeled) was incubated at 27°C for 1 h to promote nucleation, followed by 1 h of polymerization at 37°C. Polymerized microtubule samples were purified by centrifugation at 20,000$g$, 35°C for 25 min.

The glass chamber was treated as follows: first, 1 mg/ml NeutrAvidin was loaded into the chamber and incubated for 2 min to let it bind to biotin molecules covalently connected to the glass surface. The glass surface was subsequently deactivated by adding and incubating with 10× blocking solution (1 mg/ml casein and 1% pluronic F127) for 2 min. After washing them out with BRB80 containing 25% glycerol, GMPCPP-MT seeds were loaded into the chamber and incubated for 2 min to immobilize microtubules on the glass surface. Unbound microtubules were washed out with BRB80 containing 10x blocking solution and 1 mM GTP. To polymerize GDP-MT from the seeds, 28 $\mu$M Cy5-labeled tubulin (labeling rate 2%) in polymerization buffer (BRB80, 2× blocking solution, 1 mM GTP) was applied to the chamber, polymerized for 5 min, and terminated by washing free tubulin out with washout buffer (BRB80, 25% glycerol, and 2× blocking solution). To completely hydrolyze GTP, the chamber was incubated for more than 30 min before any subsequent experiments.

## Imaging

Images of in vitro assays were acquired using an Eclipse Ti (Nikon) TIRF microscope. The microscope was equipped with a 488-nm laser (Coherent OBIS488-60LS), a 532-nm laser (Coherent OBIS532-20LS), a 640-nm laser (Coherent CUBE640-40), a FF01-390/482/532/640-25 excitation filter (Semrock), a Di01-R405/488/532/635-25x36 dichroic mirror (Opto-Line), a FF01-446/510/581/703-25 fluorescence filter (Opto-Line), an Apo TIRF 60×/1.49 oil objective lens (Nikon), and a Sona sCMOS camera (Andor). All image acquisitions were managed with Micro-manager (v2.0.0; Edelstein et al, 2014). Imaging was performed at room temperature (23–26°C).

## Binding assay

D2-mCherry in imaging buffer (BRB80, 25% glycerol 1× blocking solution, 0.1% methylcellulose, 5 mM protocatechuic acid (Pacific Bioscience), 5 mM TSY (Pacific Bioscience), and 50 nM protocatechuate-3,4-dioxygenase (Pacific Bioscience)) was applied to the microtubule-immobilized glass chamber (see the Preparation for binding assay section). After 5 min of incubation, image acquisition was conducted for all registered positions. To increase the signal-to-noise ratio, five frames were taken for every position and channel.

For assays in the presence of taxol, D2-mCherry in imaging buffer was supplemented with 20 $\mu$M taxol, and images were acquired the same way. Taxol depletion was conducted by exchanging the solution with a sample of same D2 concentration without taxol. Images were taken after 5-min incubation.

Because the binding activity of D2 was lost quickly after being thawed (<1 h), we prepared new samples every 30 min.

## Binding assay with KIF5C pretreatment

During this experiment, 1 $\mu$M D2-mCherry in imaging buffer was used for the D2-mCherry samples, and variable concentrations of KIF5C in BRB30 with 25% glycerol were used for the KIF5C pretreatment. A microtubule-immobilized glass chamber (see the Preparation for binding assay section) was first incubated with D2-mCherry for 5 min. As a no pretreatment control, microtubule-bound D2 was dissociated by 1-min incubation in high-salt buffer (7.5× PEM, 25% glycerol) followed by washout with BRB80 containing 25% glycerol. Again, D2-mCherry was applied, and the chamber was incubated for 5 min before the image acquisition.

For 0.1, 0.5, and 2.0 $\mu$M KIF5C pretreatment, the incubation/dissociation cycle described below was repeated in the same chamber after the previous image acquisition of D2-bound microtubules. First, D2 was dissociated and washed out by 1-min incubation in high-salt buffer and 30-s incubation in BRB80 containing 25% glycerol. Microtubules were subsequently expanded by the addition of KIF5C and incubated for 1 min. Next, KIF5C was dissociated and washed out by a 1-min incubation in high-salt buffer and 30-s incubation in BRB80 containing 25% glycerol. Finally, D2 was applied and incubated for 5 min before the image acquisition.

For the compaction experiment, pretreatment with 2.0 $\mu$M KIF5C was repeated as above except the buffer used for the KIF5C dissociation was ADP buffer (BRB80, 25% glycerol, and 1 mM ADP) and not a high-salt buffer.

Note that because this assay takes ~1 h for each run, new D2 samples were prepared between 0 and 0.1 $\mu$M pretreatments in addition to the 0.5 and 2.0 $\mu$M pretreatments.

## Data analysis of binding assays

Images acquired by TIRF microscopy were exported as OME-TIF files. Image processing and analysis were performed using Python (v3.9.7).

To correct for uneven irradiation by the laser, images were first processed using custom site package impy (https://github.com/hanjinliu/impy; v2.1.1) as follows: For each channel, all acquired 4D image stacks (stage position, time frame, and XY) were projected along the axes of stage positions and frames by the median to

create a 2D background image. This background image was fit to a 2D diagonal Gaussian function:

$$I_{BG}(x,y) = I_0 \exp\left[-(x-\mu_x)^2/2\sigma_x^2 - (y-\mu_y)^2/2\sigma_y^2\right] + C$$

where $I_0$ is the peak background intensity, $\mu_x$ and $\mu_y$ are the peak positions, $\sigma_x$ and $\sigma_y$ are the variations of the Gaussian function, and $C$ is a constant. Subsequently, for each separate image slice, intensity values $I(x,y)$ were corrected by:

$$I'(x,y) = I(x,y)/I_{BG}(x,y) \cdot max(I_{BG}) - max(I_{BG})$$

to extract the true fluorescence intensity. The corrected image $I'(x,y)$ was averaged by the second to fifth frames and saved as 32-bit floating image stacks in TIF format. The first frame was not used for the analysis because some of them were not in focus because of the short buffering time after the stage movement.

To remove leakage of the fluorescence intensity between channels, blue-to-green, green-to-blue, red-to-green, and green-to-red leakage ratios were determined as follows. The green channel mean intensity along Alexa Fluor 488–labeled microtubules and Cy5-labeled microtubules were measured without D2-mCherry to calculate the blue-to-green and red-to-green leakage. Microtubules saturated with D2 (microtubules equilibrated with 4 $\mu$M D2 and 20 $\mu$M taxol) were used to measure the blue channel mean intensity along Cy5-labeled microtubules and red channel mean intensity along Alexa Fluor 488–labeled microtubules to calculate green-to-blue and green-to-red leakage. Leakage-corrected images were saved as 32-bit floating image stacks in TIF format.

Average fluorescence intensities along microtubules were quantified using Python image viewer napari (v0.4.16; Sofroniew et al, 2022) and custom plugin napari-filaments (https://github.com/hanjinliu/napari-filaments; v0.2.1). Briefly, fluorescence intensities (Alexa Fluor 488 channel or Cy5 channel) of the non-overlapping regions of microtubules were manually selected and fit to 2D spline curves. For the spline of length $L$, $/L/$ + 1 sample points were placed at equal intervals including the edges (i.e., placed for every ~1 pixel), and the intensities were interpolated by cubic interpolation. Average fluorescence intensities were calculated as the average of all interpolated intensities. All statistical tests were calculated using SciPy (v1.7.3), statsmodel (v0.13.2; Seabold & Perktold, 2010), or scikit-posthocs (v0.7.0; Terpilowski, 2019).

## Microtubule expansion assay

Microtubule immobilization was carried out similar to the binding assay (see the Binding assay section) but slightly modified to fit the expansion assay. After GMPCPP-MT was added, 28 $\mu$M Cy5-labeled tubulin (10% labeled) was used for a 15-min polymerization period to lengthen and brighten the microtubules. Subsequently, 8 $\mu$M Alexa Fluor 488–labeled tubulin (2% labeled) with 1 mM GMPCPP was loaded into the chamber and polymerized for 20 min. Polymerization was terminated by washing free tubulin out with washout buffer. Images were acquired before and after 4 $\mu$M D2-mCherry in imaging buffer was added to the chamber followed by the dissociation of D2 by 1-min incubation in high-salt buffer and washout by imaging buffer for 30 s.

To calculate expansion, a background correction was performed in the same way as the data analysis for the binding assay (see the Data analysis of binding assays section). Microtubule lengths were measured using napari-filaments as follows (Fig S5): first, the Cy5 channel and Alexa Fluor 488 channel of the image were added with 1:1 weight to create a total intensity image. This image was used for 2D spline fitting as mentioned in the data analysis of the binding assay (see the Data analysis of binding assays section). Next, the spline curves were clipped at the Cy5-labeled microtubule edges by fitting to the error function:

$$f(x) = \frac{a-b}{2}\left(1 + erf\left(\frac{x-x_0}{\sqrt{2}\,\sigma}\right)\right) + b$$

$$\text{where}: \quad erf(x) = \frac{2}{\sqrt{\pi}}\int_0^x e^{-t^2}dt$$

The parameters of the error function indicate that intensity increases from $b$ to $a$, where the inflection point of the intensity change is $x_0$. Cy5-labeled microtubule lengths were obtained by fragmenting the spline curve into 1,024 pieces and summing the lengths of all fragments.

## Microtubule depolymerization assay

Cy5-labeled GDP-MT (10% labeled) was grown from Alexa Fluor 488–labeled GMPCPP-MT (2% labeled) in the same way as the microtubule expansion assay (see the Microtubule expansion assay section). After 30-min incubation to hydrolyze GTP completely, the chamber was washed with washout buffer (BRB80, 25% glycerol, and 2× blocking solution) with or without 4 $\mu$M D2 and incubated for 5 min. Subsequently, glycerol-free imaging buffer (BRB80, 1× blocking solution, 0.1% methylcellulose, 5 mM protocatechuic acid, 5 mM TSY, and 50 nM protocatechuate-3,4-dioxygenase) with or without 4 $\mu$M D2 was flowed into the chamber, and the image acquisition was started immediately. For experiments without D2, images were taken at 0.5-s intervals; for those with D2, images were taken at 5-s intervals. Signals from the mCherry channel were also taken for the experiments in Fig S4.

Instead of building kymographs, we measured the decrease in Cy5 total intensity to quantify the depolymerization rate. Areas that microtubules occupied during depolymerization were manually drawn using maximum projection as the reference. The total Cy5 intensity in the area was measured for every frame, and the values were converted into length using the ratio of the microtubule length to total Cy5 intensity of the first frame. The depolymerization rate was determined by a linear regression of data points that corresponded to microtubule lengths longer than 0.5 $\mu$m using the scikit-learn package (v1.0.1; Pedregosa et al, 2011). This workflow was applied to all microtubules independently.

# Data Availability

Image data sets and Python scripts are available at https://zenodo.org/record/7634450.

# Supplementary Information

# Acknowledgements

This work is supported by JSPS KAKENHI (18K06147, 19H05379, and 21H00387 to T Shima). We thank P Karagiannis (Sofia Science Writing) for proofreading and helpful comments on the article. We also thank the members of the Uemura laboratory for valuable discussions.

## Author Contributions

H Liu: conceptualization, data curation, investigation, visualization, methodology, and writing—original draft, review, and editing.
T Shima: conceptualization, supervision, methodology, and writing—review and editing.

## Conflict of Interest Statement

The authors declare that they have no conflict of interest.

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
