## [Reviewer comments · Life Science Alliance]

Life Science Alliance

Preference of CAMSAP3 for expanded microtubule lattice contributes to stabilization of the minus end

Hanjin Liu and Tomohiro Shima

DOI: <https://doi.org/10.26508/lsa.202201714>

Corresponding author(s): Tomohiro Shima, The University of Tokyo

Review Timeline:

Submission Date:	2022-09-06
Editorial Decision:	2022-10-06
Revision Received:	2023-01-16
Editorial Decision:	2023-02-08
Revision Received:	2023-02-15
Accepted:	2023-02-15

Scientific Editor: Novella Guidi

Transaction Report:

October 6, 2022

Re: Life Science Alliance manuscript #LSA-2022-01714

Dr. Tomohiro Shima
University of Tokyo
Department of Biological Sciences
7-3-1 Hongo
Bunkyo-ku
Tokyo #113-0033
Japan

Dear Dr. Shima,

Thank you for submitting your manuscript entitled "Preference of CAMSAP3 for microtubules with expanded lattice contributes to stabilization of the minus end" to Life Science Alliance. The manuscript was assessed by expert reviewers, whose comments are appended to this letter. We invite you to submit a revised manuscript addressing the Reviewer comments.

Thank you for this interesting contribution to Life Science Alliance. We are looking forward to receiving your revised manuscript.

Sincerely,

B. MANUSCRIPT ORGANIZATION AND FORMATTING:

Reviewer #1 (Comments to the Authors (Required)):

Liu and Shima investigate the interaction of D2 domain of human CAMSAP3 with microtubule lattice. Authors use a fluorescently tagged D2 domain to investigate the localization of D2 on different microtubule lattices, including GMPCPP- and GDP-tubulin lattices, along with an expanded lattice in the presence of Kif5C. Authors report that human D2 prefers GMPCPP-microtubules over GDP-microtubules due to the expanded nature of the lattice. Authors use taxol to eliminate the effect of nucleotides on the affinity of D2 and conclude that expanded lattices due to presence of taxol is the reason for D2 localization, however, this preference is not lost following taxol depletion, except the partial D2 dissociation at low concentrations. In order to generate more physiologically relevant expanded microtubule lattices, authors use Kif5C and again observe that D2 can decorate to expanded lattices. Having established that D2 preferentially binds to expanded lattice, authors investigate whether D2 itself can expand the lattice and report that microtubule lengths increase 3% in the presence of D2. Finally, authors test whether D2 affects the microtubule depolymerization rate and show a slowdown in the presence of D2 when compared to the condition without D2. Based on their observations, authors suggest a mechanism for CAMSAP3 stretches, rather than tip tracking, in which D2 domain interacts with the microtubule lattice and allows CAMSAP3 to stay bound even in the presence of microtubule polymerization.

The experiments and figures are presented well and explained in detail within the manuscript. The data regarding the localization of D2 supports authors interpretation. The finding on D2's ability to expand the microtubule lattice is supported by the data, however, authors might include additional data from existing experiments (see comment below). On the other hand, the effect of D2 on microtubule depolymerization might require additional experiments (see my comment below). I think that further changes needed to clarify the messages in the manuscript. The following comments might be useful for authors in improving the manuscript.

- Authors mention that in order to eliminate the effect of protofilament number within GDP-tubulin extensions, authors grew them off of GMPCPP seeds. Although I agree that the technique is appropriate, I have an objection on this reasoning. The extensions from 14 PF GMPCPP seeds doesn't have to have 14 PFs, see an example in Rai et al, PNAS, 2021.

- In Figure 2C, the statistically significant differences between the taxol and depleted conditions with 250nM D2 becomes insignificant when 1uM D2 is used. Based on this, authors suggest that affinity change between D2 and microtubule is reversible at low concentrations. Although the statistics show that the intensity levels are significantly different, more concentrations needed to generalize this observation, i.e. using 100 nM and 500 nM D2. Moreover, what is authors' interpretation of the significant increase on GMPCPP-MTs versus significant decrease on GDP-MTs of DT intensity upon taxol depletion in the presence of 250nM D2?

- Based on the observations in Figure 2D, authors suggest formation of D2 clusters on GDP-MTs. However, it's hard to imagine that several micrometer long stretches can be described as clusters. Could authors justify in more detail why they described D2 stretches as clusters?

- In addition to 4C, please show length distributions of 0uM, 4uM and washed, because the existing graph doesn't show the error in determining expansion percentage.

- In Figures 5C and 5E, it would be good to show where D2 localizes, given that authors have this information already and if not could repeat the experiments with fluorescent D2. Also, it would be good to show additional frames at - for instance, 1 min, 2 min, 3 min etc - in Figure 5C. This would support the information in the kymograph shown in Figure 5E.

- Authors attribute the slowdown in the depolymerization rate to the expanded lattice due to D2. However, the design of these experiments does not allow for such conclusion being specific on the expanded lattice. The slowdown might be due to the presence of D2 on the lattice, even if it doesn't expand the lattice at all. Given that D2 leaves the lattice quickly as shown in Figure 4, authors should repeat the depolymerization assay both in the absence of D2 and glycerol (i.e. washout buffer will not include D2 and glycerol). Moreover, authors could also do a 2-step washout: First remove D2, and wait for ~90 seconds, then remove glycerol. These suggested additional experiments should take approximately the same time that it took to perform the depolymerization assay that has been already included.

- In Figure 6, I ask for few clarifications in A(3). Do authors mean that CAMSAP3 remains bound on the microtubule even after microtubule is polymerized due to the strong interaction between D2 domain and the expanded microtubule lattice? "Capture" would mean that additional molecules come and bind, not the existing ones stay bound. Also please use a legend for expanded lattice and compact lattice. In a more general view regarding Figure 6, what about the idea of GTP hydrolysis and lattice compaction as a result of it? Authors do not mention in detail the differences between the GTP-cap and GDP-lattice in terms of the compaction as for example Zhang et al. PNAS, 2018 reports. What authors draw, specifically in 6B, assumes a compacted lattice as well as compacted ends based on their color scheme. In fact, authors use glycerol in their assay and wait for hydrolysis to complete in their experiments. However, to mention polymerization and end tracking ability of CAMSAPs, authors need to consider the GTP-cap and GDP-lattice differences. If the CKK domain is binding at the very end of the microtubules which are not hydrolyzed yet, can we even speak of lattice expansion by D2 domain? Do authors assume that only the terminal dimers are not hydrolyzed, but others in the lattice starting from the 2nd dimer along the protofilament are hydrolyzed, therefore compacted? Based on the earlier work on the localization of EB family proteins, this assumption should be wrong to begin with. Although the majority of EB localization work is done on microtubule plus ends, Strothman et al. JCB, 2019 has shown that EB localization on the minus end shows a similar trend as the plus ends.

- Please put legend for colors on Figures 1B, 3A, 4A, 4C.

Reviewer #2 (Comments to the Authors (Required)):

CAMSAPs are microtubule-associated proteins that bind to microtubule minus-ends, prevent catastrophe, and have proposed roles in non-centrosomal microtubule nucleation. Here, Liu and Shima use purified, *in vitro* assays to show that the D2 domain of CAMSAP3 preferentially binds to expanded microtubule lattices and that D2 binding expands compacted GDP-bound microtubule lattices. The experiments strongly demonstrate that D2 localizes to microtubule lattices expanded via the non-hydrolyzable GTP analogue GMPCPP, the microtubule-targeting drug taxol, or by the microtubule motor KIF5C. These findings provide insights into the molecular mechanism through which CAMSAPs recognize and stabilize microtubule minus ends.

However, all experiments are performed with the D2 domain alone and the role of the D2 domain in the context of full-length CAMSAP3 is unclear. D2 is located adjacent to the CKK domain, which has been previously shown to confer preferential binding of CAMSAP to microtubule minus ends (Atherton et al., 2017, 2019). These results suggest that the CKK domain alone might also sense and/or potentiate lattice expansion. Alternatively, CKK might sense different structural elements like protofilament spacing. Thus, it is unclear if in the full-length protein, CKK would strengthen or weaken lattice expansion by D2. Experiments comparing the expansion of constructs containing both the CKK and D2 domain or the two domains alone would increase the impact of the manuscript by allowing the authors to better test their proposed model.

The authors propose a model where CKK initiates the initial binding of CAMSAP to the microtubule minus end, leading to a local enrichment of the D2 domain sufficient to expand the microtubule lattice. Yet, the rationale for the hypothesis that the minus-end is initially in a compacted state is unclear. It seems likely that association with nucleating factors, possibly including CAMSAP, would potentially hold the minus end in an expanded state. This idea is consistent with the taxol washout experiments, where D2 remains stably bound to the expanded lattice even in the absence of taxol.

Minor points (in order of importance):

- In the taxol depletion experiments (Fig. 2), is it possible that the D2-mcherry is stabilizing the expanded lattice and preventing taxol from unbinding? The authors should repeat the assay by washing with a high-salt buffer devoid of taxol to remove both taxol and D2, then replace with D2-containing buffer. Alternatively, they could perform assays at lower [D2] where effects of stabilization would be expected to be weaker.
- In the depolymerization assay (Fig. 5), D2 at 4 μ M binds the both expanded and compacted microtubule lattice (Fig. 1). Does D2 alter MT dynamics at lower concentrations where it specifically binds to the minus end?
- In Fig. 1, the authors should clarify that taxol is not present and that microtubules were stabilized by 25% glycerol in the figure legend.
- On p. 1, paragraph 1: The authors might consider revising "Microtubules are a eukaryotic cytoskeleton ..." to "Microtubules are a eukaryotic cytoskeletal polymer ..." or "Microtubules are a component of the microtubule cytoskeleton ..."

We greatly appreciate both reviewers for their constructive and useful comments and suggestions, which helped us to improve the manuscript. As indicated in the responses below, we have taken all these comments and suggestions into account in preparing the revised manuscript. The changes in the revised manuscript were marked yellow and our responses below were colored blue.

Reviewer #1 (Comments to the Authors (Required)):

- Authors mention that in order to eliminate the effect of protofilament number within GDP-tubulin extensions, authors grew them off of GMPCPP seeds. Although I agree that the technique is appropriate, I have an objection on this reasoning. The extensions from 14 PF GMPCPP seeds doesn't have to have 14 PFs, see an example in Rai et al, PNAS, 2021.

Thank you for your suggestion. We have rewritten the reasoning of the use of GMPCPP seeds (p. 6, line 126-130).

- In Figure 2C, the statistically significant differences between the taxol and depleted conditions with 250nM D2 becomes insignificant when 1uM D2 is used. Based on this, authors suggest that affinity change between D2 and microtubule is reversible at low concentrations. Although the statistics show that the intensity levels are significantly different, more concentrations needed to generalize this observation, i.e. using 100 nM and 500 nM D2. Moreover, what is authors' interpretation of the significant increase on GMPCPP-MTs versus significant decrease on GDP-MTs of DT intensity upon taxol depletion in the presence of 250nM D2?

As suggested, we repeated the same assay with 100 nM D2 and added the data to Fig. 2. The results were consistent with the trends shown in the data with 250 nM D2, supporting our suggestion (p.7 line 153, 168-177).

We also added our interpretation for the different directions of change in fluorescence intensity by taxol-depletion depending on the type of microtubule and D2 concentration in the Discussion part (p.11 line 325-p. 12 line 337).

- Based on the observations in Figure 2D, authors suggest formation of D2 clusters on GDP-MTs. However, it's hard to imagine that several micrometer long stretches can be described as clusters. Could authors justify in more detail why they described D2 stretches as clusters?

We had described D2 stretches as clusters to emphasize their distinct affinity with D2 in the regions over other GDP-MT regions. However, as you pointed out, the term "cluster" usually refers to objects that are smaller in size. Therefore, we revised to clusters as subregions of GDP-MT. Based on our results, we assume that the existence of D2 stretches suggests positive feedback between D2 dissociation and MT lattice compaction. We have added this discussion to the manuscript (p. 11 line 314-322).

- In addition to 4C, please show length distributions of 0uM, 4uM and washed, because the existing graph doesn't show the error in determining expansion percentage.

We added Fig. 4D to show the microtubule length distribution in the 0 μ M condition and length independency of the expansion percentages.

- In Figures 5C and 5E, it would be good to show where D2 localizes, given that authors have this information already and if not could repeat the experiments with fluorescent D2. Also, it would be good to show additional frames at – for instance, 1 min, 2 min, 3 min etc – in Figure 5C. This would support the information in the kymograph shown in Figure 5E.

We repeated the experiments. We confirmed that D2 does not accumulate to any parts of depolymerizing microtubules by averaging line scans of microtubule tips (Fig. S4, p.9 line 250-254). We also added more frames in Fig. 5E.

- Authors attribute the slowdown in the depolymerization rate to the expanded lattice due to D2. However, the design of these experiments does not allow for such conclusion being specific on the expanded lattice. The slowdown might be due to the presence of D2 on the lattice, even if it doesn't expand the lattice at all. Given that D2 leaves the lattice quickly as shown in Figure 4, authors should repeat the depolymerization assay both in the absence of D2 and glycerol (i.e. washout buffer will not include D2 and glycerol). Moreover, authors could also do a 2-step washout: First remove D2, and wait for ~90 seconds, then remove glycerol. These suggested additional experiments should take approximately the same time that it took to perform the depolymerization assay that has been already included.

Because D2-triggered lattice expansion was canceled immediately after D2 dissociation from microtubules (Fig. 4, p.9 line 228-229), it is hard to prepare expanded, non-D2-decorated microtubule. With the suggested experiments, although D2 dissociated within ~90 seconds, microtubule lattice also reverted to the compacted state. If we make a D2 mutant that decouple its MT binding and lattice expansion abilities, the question would be accessible, but we believe that kind of experiments should be included in future not the present study.

- In Figure 6, I ask for few clarifications in A(3). Do authors mean that CAMSAP3 remains bound on the microtubule even after microtubule is polymerized due to the strong interaction between D2 domain and the expanded microtubule lattice? "Capture" would mean that additional molecules come and bind, not the existing ones stay bound. Also please use a legend for expanded lattice and compact lattice. In a more general view regarding Figure 6, what about the idea of GTP hydrolysis and lattice compaction as a result of it? Authors do not mention in detail the differences between the GTP-cap and GDP-lattice in terms of the compaction as for example Zhang et al. PNAS, 2018 reports. What authors draw, specifically in 6B, assumes a compacted lattice as well as compacted ends based on their color scheme. In fact, authors use glycerol in their assay and wait for hydrolysis to complete in their experiments. However, to mention polymerization and end tracking ability of CAMSAPs, authors need to consider the GTP-cap and GDP-lattice differences. If the CKK domain is binding at the very end of the microtubules which are not hydrolyzed yet, can we even speak of lattice expansion by D2 domain? Do authors assume that only the terminal dimers are not hydrolyzed, but others in the lattice starting from the 2nd dimer along the protofilament are hydrolyzed, therefore compacted? Based on the earlier work on the localization of EB family proteins, this assumption should be wrong to begin with. Although the majority of EB localization work is done on microtubule plus ends, Strothman et al. JCB, 2019 has shown that EB localization on the minus end shows a similar trend as the plus ends.

We modified the manuscript not to use “capture” and added description of lattice states in Fig. 6.

There are two reasons that we excluded the effect of GTP cap from our model. First, GTP-cap is not expanded in such way that D2 prefers. In Atherton et al., *Nat Struct Mol Biol.*, 2017, Fig. S1B, no D2 intensity is visible at either end. Similarly, in Zhang et al., *eLife*, 2017, Fig. 2C, TPX2^{micro} (the minimal TPX2 construct that expands microtubule and shows GMPCPP preference) did not show end tracking. Second, even if a GTP-cap is expanded, proteins like D2 that can keep the expansion, or in other words, make the expansion more long-lived, are required to make the microtubule end more stable than the intact form. We added this to the discussion part (p.10 line 263-265, p. 12 line 347-349).

- Please put legend for colors on Figures 1B, 3A, 4A, 4C.

The figure legends are modified as indicated.

Reviewer #2 (Comments to the Authors (Required)):

CAMSAPs are microtubule-associated proteins that bind to microtubule minus-ends, prevent catastrophe, and have proposed roles in non-centrosomal microtubule nucleation. Here, Liu and Shima use purified, in vitro assays to show that the D2 domain of CAMSAP3 preferentially binds to expanded microtubule lattices and that D2 binding expands compacted GDP-bound microtubule lattices. The experiments strongly demonstrate that D2 localizes to microtubule lattices expanded via the non-hydrolyzable GTP analogue GMPCPP, the microtubule-targeting drug taxol, or by the microtubule motor KIF5C. These findings provide insights into the molecular mechanism through which CAMSAPs recognize and stabilize microtubule minus ends.

However, all experiments are performed with the D2 domain alone and the role of the D2 domain in the context of full-length CAMSAP3 is unclear. D2 is located adjacent to the CKK domain, which has been previously shown to confer preferential binding of CAMSAP to microtubule minus ends (Atherton et al., 2017, 2019). These results suggest that the CKK domain alone might also sense and/or potentiate lattice expansion. Alternatively, CKK might sense different structural elements like protofilament spacing. Thus, it is unclear if in the full-length protein, CKK would strengthen or weaken lattice expansion by D2. Experiments comparing the expansion of constructs containing both the CKK and D2 domain or the two domains alone would increase the impact of the manuscript by allowing the authors to better test their proposed model.

We prepared a D2-CKK construct as suggested. The construct showed ~1% of lattice expansion, which is lower than the expansion shown by D2 alone in the same condition (~3%). The result suggests that D2, not CKK, primarily leads to the lattice expansion. We have added the result as Fig. S2 and a description about the experiment in the revised manuscript (p. 8 line 220-222).

The authors propose a model where CKK initiates the initial binding of CAMSAP to the microtubule minus end, leading to a local enrichment of the D2 domain sufficient to expand the microtubule lattice. Yet, the rationale for the hypothesis that the minus-end is initially in a compacted state is unclear. It seems likely that association with nucleating factors, possibly including CAMSAP, would potentially hold the minus end in an expanded state. This idea is consistent with the taxol washout experiments, where D2 remains stably bound to the expanded lattice even in the absence of taxol.

As you pointed out, the polymerizing tip of microtubules is expected to have an expanded lattice; however, D2 does not preferentially bind to that region. In Atherton et al., Nat Struct Mol Biol., 2017, Fig. S1B, no D2 intensity is visible at either microtubule end. Similarly, in Zhang et al., eLife, 2017, Fig. 2C, TPX2^{micro} (the minimal TPX2 construct that expands microtubule and shows preference to GMPCPP-MTs) did not show end tracking. Therefore, the lattice at the polymerizing tips should differ from those of the expanded lattices tested here (GMPCPP-, KIF5-pretreated and taxol-stabilized MTs) at least in terms of affinity to D2. We added this explanation in the discussion section (p.10 line 263-265, p. 12 line 347-349).

Minor points (in order of importance):

- In the taxol depletion experiments (Fig. 2), is it possible that the D2-mcherry is stabilizing the expanded lattice

and preventing taxol from unbinding? The authors should repeat the assay by washing with a high-salt buffer devoid of taxol to remove both taxol and D2, then replace with D2-containing buffer. Alternatively, they could perform assays at lower [D2] where effects of stabilization would be expected to be weaker.

We repeated the assay at 100 nM D2 concentration (Fig. 2 and Table 1), and the results were totally consistent with the notion that the effect of taxol is reversible.

- In the depolymerization assay (Fig. 5), D2 at 4 μ M binds the both expanded and compacted microtubule lattice (Fig. 1). Does D2 alter MT dynamics at lower concentrations where it specifically binds to the minus end?

D2 alone does not specifically bind to microtubule minus ends at any concentration, since minus-end binding is a function of the CKK domain. As mentioned in discussion, when a CAMSAP3 molecule bound at the minus end of microtubule, the local concentration of D2 is expected to be $>20 \mu$ M. Therefore, it is hard to examine the effects of D2 specifically located at the minus end at lower concentrations.

- In Fig. 1, the authors should clarify that taxol is not present and that microtubules were stabilized by 25% glycerol in the figure legend.

The legend was modified as indicated.

- On p. 1, paragraph 1: The authors might consider revising "Microtubules are a eukaryotic cytoskeleton ..." to "Microtubules are a eukaryotic cytoskeletal polymer ..." or "Microtubules are a component of the microtubule cytoskeleton ..."

We revised the description as suggested.

February 8, 2023

RE: Life Science Alliance Manuscript #LSA-2022-01714R

Dr. Tomohiro Shima
The University of Tokyo
Department of Biological Sciences
7-3-1 Hongo
Bunkyo-ku
Tokyo #113-0033
Japan

Dear Dr. Shima,

Thank you for submitting your revised manuscript entitled "Preference of CAMSAP3 for expanded microtubule lattice contributes to stabilization of the minus end". We would be happy to publish your paper in Life Science Alliance pending final revisions necessary to meet our formatting guidelines.

- please address Reviewer 2's remaining question
- please add the Twitter handle of your host institute/organization as well as your own or/and one of the authors in our system
- please use the [10 author names, et al.] format in your references (i.e. limit the author names to the first 10)
- please include a Figure Legends section in your main manuscript text

A. FINAL FILES:

B. MANUSCRIPT ORGANIZATION AND FORMATTING:

**Submission of a paper that does not conform to Life Science Alliance guidelines will delay the acceptance of your

manuscript.**

The license to publish form must be signed before your manuscript can be sent to production. A link to the electronic license to publish form will be sent to the corresponding author only. Please take a moment to check your funder requirements.

Sincerely,

Reviewer #1 (Comments to the Authors (Required)):

Liu and Shima provided a revised manuscript which is about investigating the interaction of CAMSAP3-D2 domain with the microtubule lattice. Authors report that D2 domain identifies the expanded microtubule lattice. D2 additionally is able to expand the compacted microtubule lattice. Based on these observations, authors suggest that CAMSAP has both reading and writing ability in regards to the microtubule lattice structure. In my opinion, authors nicely present their data in the figures and their interpretations will provide detailed knowledge on the interaction between microtubules and CAMSAP. In the revised manuscript, authors incorporated the suggestions from the reviewers. In the rebuttal letter, authors also explain well why they cannot incorporate few of the suggestions in this manuscript. The rebuttal letter was satisfactory. Based on the revision and the rebuttal letter, I now recommend the publication of this manuscript.

Reviewer #2 (Comments to the Authors (Required)):

The authors have performed substantial new experiments and analysis to address the points raised in the review. They purified a construct containing both the D2 and CKK domains, and show that the D2-CKK construct expands the lattice by 1% compared to 3% for the D2 domain alone. This result supports their hypothesis that the D2 domain is responsible for lattice expansion. It would be helpful for the authors to clarify in the manuscript: is the reduced expansion due to preferential localization of D2-CKK at the microtubule minus end, compared to D2 which decorates the entire lattice? Stated in another way -- does the D2-CKK domain also strongly expand the lattice, but only where it is enriched at the minus end?

The additional data and clarifications to the text addressed my other concerns.

We greatly appreciate both reviewers for their careful and constructive review on our manuscript. Below shows our response (in blue) to the Reviewer 2's remaining question.

Reviewer #2 (Comments to the Authors (Required)):

The authors have performed substantial new experiments and analysis to address the points raised in the review. They purified a construct containing both the D2 and CKK domains, and show that the D2-CKK construct expands the lattice by 1% compared to 3% for the D2 domain alone. This result supports their hypothesis that the D2 domain is responsible for lattice expansion. It would be helpful for the authors to clarify in the manuscript: is the reduced expansion due to preferential localization of D2-CKK at the microtubule minus end, compared to D2 which decorates the entire lattice? Stated in another way -- does the D2-CKK domain also strongly expand the lattice, but only where it is enriched at the minus end?

At 4 μ M D2-CKK, the construct bound along the entire length of the microtubules, as did D2. Also, the fluorescence intensity of D2-CKK and D2 on microtubules showed no significant difference. These results suggest that the difference in the expansion rate of D2 and D2-CKK reflects their expansion ability and is not due to the difference in their localization along microtubules. We have added this description on line 220-222, p. 8, Fig. S2D and S2E.

February 15, 2023

RE: Life Science Alliance Manuscript #LSA-2022-01714RR

Dr. Tomohiro Shima
The University of Tokyo
Department of Biological Sciences
7-3-1 Hongo
Bunkyo-ku
Tokyo #113-0033
Japan

Dear Dr. Shima,

Thank you for submitting your Research Article entitled "Preference of CAMSAP3 for expanded microtubule lattice contributes to stabilization of the minus end". It is a pleasure to let you know that your manuscript is now accepted for publication in Life Science Alliance. Congratulations on this interesting work.

DISTRIBUTION OF MATERIALS:

Again, congratulations on a very nice paper. I hope you found the review process to be constructive and are pleased with how the manuscript was handled editorially. We look forward to future exciting submissions from your lab.

Sincerely,
